

# The influence of transformed Reynolds number limitation on gas transfer parameterizations and global DMS and CO$_2$ fluxes

Alexander Zavarsky[1] and Christa A. Marandino[2]

[1]Researcher, Kiel, Germany
[2]GEOMAR Helmholtz Centre for Ocean Research, Kiel, Germany

**Correspondence:** Alexander Zavarsky (alexz@mailbox.org)

**Abstract.** Eddy covariance measurements show gas transfer velocity limitation at medium to high wind speed. A wind-wave interaction described by the transformed Reynolds number is used to characterize environmental conditions favoring this limitation. We take the transformed Reynolds number parameterization to review the two most cited wind speed gas transfer velocity parameterizations, Nightingale 2000 and Wanninkhof 1992/2014. We propose an algorithm to correct for the effect of
gas transfer limitation and validate it with two gas transfer limited directly measured DMS gas transfer velocity data sets. A correction of the Nightingale 2000 parameterization leads to an average increase of 22 % of its predicted gas transfer velocity. The increase for Wanninkhof 2014 is 9.85 %. Additionally, the correction is applied to global air-sea flux climatologies of CO$_2$ and DMS. The global application of gas transfer limitation leads to a decrease of 6-7 % for the uptake CO$_2$ by the oceans and to decrease of 11 % of oceanic outgassing of DMS. We expect the magnitude of Reynolds limitation on any global air-sea gas
exchange to be about 10 %.

## 1   Introduction

Gas transfer F between the ocean and the atmosphere is commonly described as the product of the concentration difference $\Delta C$ between the liquid phase (seawater) and the gas phase (atmosphere) and the gas transfer velocity k. $\Delta C$ acts as the forcing potential difference and k as the conductance, which includes all processes promoting and limiting gas transfer. $c_{air}$ and $c_{water}$
are the respective air-side and water-side concentrations. H is the dimensionless form of Henry's law constant.

$$F = k \cdot \Delta C = k \cdot \left( c_{air} - \frac{c_{water}}{H} \right) \tag{1}$$

$\Delta C$ is typically measured with established techniques, although the distance of the measurements from the interface introduce uncertainties in the flux calculation. Parameterizations of k are another source of uncertainty in calculating fluxes. The flux F can be directly measured, for example with the eddy covariance technique, together with $\Delta C$ in order to derive k and estimate
a k parameterization (Eq. (2)).

$$k = \frac{F}{\Delta C} = \frac{F}{c_{air} - \frac{c_{water}}{H}} \tag{2}$$

It is very common that k is parameterized with wind speed and all wind speed parameterizations have in common that k increases monotonically with increasing wind speed. This assumption is sensible, as higher wind speed increases turbulence



both on the air and the water side and hence the flux. Additional processes like bubble generation can additionally enhance gas transfer. The total gas transfer velocity $k_{total}$, which is measured by eddy covariance or other direct flux methods, is split up into the water side gas transfer velocity $k_{water}$ and the air side gas transfer velocity $k_{air}$ (Eq. (3)).

$$\frac{1}{k_{total}} = \frac{1}{k_{water}} + \frac{1}{k_{air}} \tag{3}$$

We focus, in this work, on $k_{water}$ which is the sum of the interfacial gas transfer $k_o$ and the bubble mediated gas transfer $k_b$ (Eq. (4)).

$$k_{water} = k_o + k_b \tag{4}$$

To make gas transfer velocities of different gases comparable, Schmidt number (Sc) (Eq. (5)) scaling has been introduced. Sc scaling only applies to $k_o$ and $k_{air}$. Sc is the ratio of the viscosity $\nu$ to the diffusivity D of the respective gas in seawater.

$$Sc = \frac{\nu}{D} \tag{5}$$

$$\frac{k_{o,Sc}}{k_{o,660}} = \left(\frac{Sc}{660}\right)^n \tag{6}$$

The exponent n is chosen depending on the surface properties. For smooth surfaces $n=-\frac{2}{3}$ and rough wavy surfaces $n=-\frac{1}{2}$ (Komori et al., 2011). In this study $n=-\frac{1}{2}$ is used.

In contrast to commonly accepted gas transfer velocity parameterizations, parameterizations based on direct flux measurements by eddy covariance systems have shown a decrease or flattening of k with increasing wind speed at medium to high wind speed (Bell et al., 2013, 2015; Yang et al., 2016; Blomquist et al., 2017; Zavarsky et al., 2018).

Here we use the transformed Reynolds number $Re_{tr}$(Zavarsky et al., 2018) to identify instances of gas transfer limitations.

$$Re_{tr} = \frac{u_{tr} \cdot H_s}{\nu_{air}} \cdot cos(\theta) \tag{7}$$

$Re_{tr}$ is the Reynolds number transformed into the reference system of the moving wave. $u_{tr}$ is the wind speed relative to the wave, $H_s$, the significant wave height, $\nu_{air}$ the kinematic viscosity of air and $\theta$ the angle between the wave crest and the transformed wind direction. A flux measurement at values of $Re_{tr} \leq 6.7 \cdot 10^5$ is gas transfer limited(Zavarsky et al., 2018). This parameterization shows that the limitation is primarily dependent on wind speed, wave speed, wave height and a directional component. It is noteworthy that so far only eddy covariance deduced gas transfer velocities have shown a gas transfer

limitation. This may be due to the spatial (1 km) and temporal (30 min) resolution of EC measurements, or to the types of gases measured (e.g. $CO_2$, DMS, OVOCs). The use of rather soluble gases (DMS, acetone, methanol) makes the gas transfer velocity not greatly influenced by bubble mediated gas transfer. Gas transfer limitation only affects $k_o$(Zavarsky et al., 2018). Another direct flux measurement technique, the dual tracer method, utilizes sulphur hexafluoride ($SF_6$) or $^3$He. The dual tracer measurement usually lasts over few days but could have a similar spatial resolution as eddy covariance. $SF_6$ and $^3$He are both

very insoluble and heavily influenced by the bubble effect. Hence, if the gas transfer limitation only affects $k_o$, $k_b$ could be





masking the gas transfer limitation. Additionally, the long measurement period could decrease the likelihood of detection of gas transfer limitation as the conditions for limitation might not be persistent over a few days.

Using wind and wave data for the year 2014, we calculate $Re_{tr}$ and perform an analysis of the impact of gas transfer limitation on the yearly global air sea exchange of $CO_2$ and DMS. So far global estimates of air-sea exchange of these two gases(Lana

et al., 2011; Takahashi et al., 2009; Rödenbeck et al., 2015) have been based on k parameterization which have not included a mechanism for gas transfer limitation. We provide an iterative calculation of the effect of gas transfer limitation and apply the correction to existing $CO_2$ and DMS climatologies.

We investigate the two most commonly used gas parameterizations (both cited more than 1000 times each) for the occurrence of gas transfer limitation. The Nightingale 2000 (N00)(Nightingale et al., 2000) parameterization contains data from the

North Sea, Florida Strait and the Georges Bank between 1989-1996. N00 derived the gas transfer velocity from changes in the ratio of $SF^6$ and $^3He$ (dual trace method). We also compare N00 to the gas transfer parameterization Wanninkhof 2014 (W14)(Wanninkhof, 2014) which is an update to Wanninkhof 1992(Wanninkhof, 1992). They use natural and anthropogenically produced carbon isotopes to estimate the air-sea flux over several years. Using a wind speed climatology they can deduce a quadratic k vs wind speed parameterization. The parameterization W14 must already have gas transfer limitation included

as it is solely dependent on seawater measurements of carbon isotopes. The gas transfer limitation is averaged as they use a global, multi-year approach. All studied k vs u relationships (N00,W14) are monotonically increasing with wind speed.

## 2 Methods

### 2.1 Wave Watch Model III

We use wave data from the WWIII model hindcast run by the Marine Modeling and Analysis Branch of the Environmental

Modelling Center of the National Center for Environmental Prediction (NCEP)(Tolman, 1997, 1999, 2009). The data was obtained for the total year 2014 with a temporal resolution of 3 hours and a spatial resolution of 0.5° x 0.5°. It also provides the u (meridional) and v (zonal) wind vectors, assimilated from the Global Forecast System, used in the model. We retrieved wind speed, wind direction, bathymetry, wave direction, wave period and significant wave height. We converted the wave period $T_p$ to phase speed $c_p$, assuming deep water waves, using Eq. (8)(Hanley et al., 2010).

$$c_p = \frac{g \cdot T_p}{2\pi} \tag{8}$$

### 2.2 Auxiliary variables

Surface air temperature T, air pressure p, sea surface temperature SST and sea ice concentration were retrieved from the ERA-Interim reanalysis of the European Center for Meridional Weather Forecast(Dee et al., 2011). It provides a six hourly resolution and a global $0.125^o$ x $0.125^o$ spatial resolution. Sea surface salinity (SSS) was extracted from the Takahashi clima-

tology(Takahashi et al., 2009).

Air-sea partial pressure difference ($\Delta pCO_2$) was obtained from the Takahashi climatology. $\Delta pCO_2$, in the Takahashi climato-





logy, is calculated for the year 2000 $CO_2$ air concentrations. Assuming an increase in both the air concentration and the partial pressure in the water side, the partial pressure difference remains constant. The dataset has a monthly temporal resolution, a $4^o$ latitudinal resolution and a $5^o$ longitudinal resolution.

DMS water concentrations were taken from the Lana DMS climatology (Lana11)(Lana et al., 2011). These are provided with a monthly resolution and a $1^o$ x $1^o$ spatial resolution. The air mixing ratio of DMS was set to zero $c_{air,DMS} = 0$. Taking air mixing ratios into account, the global air sea flux of DMS reduces by 17 %(Lennartz et al., 2015). We still think that our approach is reasonable, as we are looking at the change of flux due to gas transfer limitation only.

We linearly interpolated all datasets to the grid and times of the WWIII model.

## 2.3 Kinematic viscosity

The kinematic viscosity $\nu$ of air is dependent on air's density $\rho$ and the dynamic viscosity $\mu$ of air, Eq. (9).

$$\nu(T,p) = \frac{\mu(T)}{\rho(T,p)} \tag{9}$$

The dynamic viscosity is dependent on temperature T and can be calculated using Sutherland's law(White, 1991) (Eq. (10)).

$$\mu = \mu_0 \cdot \left(\frac{T}{T_0}\right)^{\frac{2}{3}} \tag{10}$$

$\mu_0 = 1.716 \cdot 10^{-5}$ N s m$^{-2}$ at $T_0 = 273$ K(White, 1991). Air density is dependent on temperature T and air pressure p and was calculated using the ideal gas law.

## 2.4 Transformed Reynolds number

The Reynolds number describes the balance of inertial forces and viscous forces. It is the ratio of the typical length and velocity scale over the kinematic viscosity. The transformed Reynolds number, in Eq. (11), uses the wind speed $u_{tr}$, transformed into the wave's reference system. The significant wave height $H_s$ is used as the typical length scale. The difference between wind direction and wave direction is given by the angle $\theta$. The factor $cos(\theta)$ is multiplied to $H_s$ to account for directional dependencies. Wind at an angle of $\theta = 90^o$, for example, does not experience a wave crest or trough, but rather an along-wind corrugated surface

$$Re_{tr} = \frac{u_{tr} \cdot H_s}{\nu} \cdot cos(\theta) \tag{11}$$

## 3 Gas transfer limitation model

Below $Re_{tr} \leq 6.7 \cdot 10^5$ a flow separation between the sea surface and the wind flowing above the wave limits gas transfer(Zavarsky et al., 2018). As a result, common wind speed parameterizations of k are not applicable (Eq. (1)). To provide a magnitude of this limitation we propose an alternative wind speed $u_{alt}$, which is lower than $u_{10}$. This decrease accounts for the effect of gas transfer limitation. $u_{alt}$ can then be used with k parameterizations to calculate the gas flux.



Given a set wave field, if the relative wind speed in the reference system of the wave $u_{tr}$ is big enough that the transformed Reynolds number is greater than the threshold of $6.7 \cdot 10^5$, no limitation occurs. In the no limitation case, k can be estimated by common gas transfer parameterizations. If the wind speed is getting close to the wave's phase speed, the transformed Reynolds number drops below the threshold, flow separation happens, and limitation occurs. We propose, to estimate the magnitude of

the limitation, a stepwise reduction of $u_{10}$. We recalculated $Re_{tr}$ with a lower $u_{alt} \prec u_{10}$ and decrease $u_{alt}$ as long as $Re_{tr}$ is below the threshold. If the $Re_{tr}$ crosses to the non limiting regime the actual $u_{alt}$ can be used as an alternative wind speed. The iteration steps are: [1] Calculate $Re_{tr}$ and check if $Re_{tr} \le 6.7 \cdot 10^5$. [2] If yes, reduce the wind speed $u_{10}$ in the earth reference system to $u_{alt}$ and recalculate $Re_{tr}$ and check if $Re_{tr} \le 6.7 \cdot 10^5$. [3] If $Re_{tr}$ is greater than the threshold stop. Then the new reduced wind speed $u_{alt}$ can be used in the k parameterization, if otherwise continue with step [1]. The step size in

this model was 0.3 m s$^{-1}$. We think that this is a good balance of computing time and and velocity resolution of the step size. The minimum velocity for $u_{alt}$ is 0 m s$^{-1}$. Figure 1 shows a flowchart of the algorithm. This algorithm is applied to every box at every time step.

A change in the parameters of the wave field is, in our opinion, not feasible as the wave field is externally prescribed. Swell travels long distances and does not necessarily have a direct relation to the wind conditions at the location of the gas transfer

and measurement. Therefore, we change the wind speed only.

## 3.1 Gas transfer

For the global air-sea exchange of DMS and $CO_2$ we use the bulk gas transfer formula (Eq. (1)). We calculate for every grid box and every time step $u_{alt}$ according to the description in Sect. 3. If $u_{alt}$ is lower than $u_{10}$ from the global reanalysis then

gas transfer limitation occurs and $u_{alt}$ is used in the bulk gas transfer formula (Eq. (1)). The difference between $u_{alt}$ and $u_{10}$ directly relates to the magnitude of gas transfer limitation.

We assume that the gas transfer limitation only affects $k_o$. Therefore only the parameterization of $k_o$ should be altered using the new reduced wind speed $u_{alt}$. This is especially problematic for rather insoluble gases with a high contribution of bubble mediated gas transfer, like $CO_2$ (at high wind speed), $SF_6$, [3]He. We use a linear parameterization, ZA18(Zavarsky et al., 2018),

and the quadratic parameterizations, Tak09(Takahashi et al., 2009), W14 and NI00. The ZA18 parameterization is a linear fit to all data points $< 10$ m s$^{-1}$ obtained during a cruise in the Indian Ocean (SO234-2/235). This linear fit does not contain data points which are influenced by gas transfer limitation. $u_{alt}$ can be directly inserted into ZA18 as we do not expect a large bubble contribution to k (Eq. (12)), because of the solubility of DMS. However, all other parameterizations are based on measurements with rather insoluble gases, which have a significant bubble mediated gas transfer contribution. As a consequence we subtract

a linear dependency using the ZA18 parametrization, to account for the gas transfer limitation in $k_o$ (Eq. (13)).

$$F_{lim,ZA18} = (3.1 \cdot u_{alt} - 5.37) \cdot \Delta C \tag{12}$$

$$F_{lim,Tak00/W14/NI00} = \left( k_{Tak00/W14/NI00}(u_{10}) - k_{ZA18}(u_{alt}) \right) \cdot \Delta C \tag{13}$$



For the global DMS transfer we use ZA18 and NI00, which is also used by Lana11(Lana et al., 2011). We parameterize the $CO_2$ flux using Tak09, NI00 and W14.

Sea ice concentration from the ERA-Interim reanalysis was included as a linear factor in the calculation. A sea ice concentration of 90 %, for example, results in a 90 % reduction of the flux. Each time step (3 h) of the WWIII model provided a global grid

of air-sea fluxes with and without gas transfer limitation. Theses single time steps were summed up to get a yearly flux result. For $CO_2$, $\Delta C$ was directly provided by the Takahashi climatology. For DMS the air concentration was neglected and $\Delta C$ reduced to $c_{water}$, which was obtained from Lana11.

## 4    Results

We apply the correction to two data sets (Knorr11 (Bell et al., 2017) and SO234-2/235(Zavarsky et al., 2018)) of DMS gas

transfer velocities. Both data sets experienced gas transfer limitation at high wind speed. Using this proof of concept, we quantify the influence of gas transfer limitation on N00 and W14 and correct for it. Finally, we apply the correction to global flux estimates of $CO_2$ and DMS.

### 4.1    Correction of the interfacial gas transfer

Figures 2 and 3 show the corrected DMS gas transfer velocities for the SO234-2/235 and the Knorr11 cruises. The black circles

indicate the original data set. The colored circles are k values at the corrected wind speed. If a black circle and a colored circle are concentric the data point was not limited and therefore no correction was applied. For comparison, in both Fig., the linear fit to the data SO234-2/235 below 10 m s$^{-1}$ (ZAV17) is plotted. Both Fig.s show the significant wave height as color.

Figure 2 shows the gas transfer limited data points at 14-16 m s$^{-1}$ moved closer to the linear fit after correction to $u_{alt}$. The linear fits to the data set before and after the correction are shown with a dotted (before) and dashed (after) line. The large gas

transfer velocity values at around 13 m s$^{-1}$ and above 35 cm h$^{-1}$ were moved to 11 m$^{-1}$. This means a worsening of the the k estimate by the linear fit. These data points have very low $\Delta C$ values(Zavarsky et al., 2018), therefore, we expect a large scatter as a result from Eq. (2).

Figure 3 also shows an improvement of the linear fit estimates. The gas transfer limited data points were assigned the new wind speed $u_{alt}$, resulting in better agreement with the linear fit of SO234-2/235. The change of the linear fit to the corrected and

uncorrected data set can be seen in the dotted (before) and dashed (after) line. The corrected data points at 12-16 m s$^{-1}$ are still, relative to the linear estimates, heavily gas transfer limited. A reason could be that the significant wave height of these points is larger than 3.5 m and they experienced high wind speed. A shielding of wind by the large wave or an influence of water droplets on the momentum transfer is suggested as reason(Yang et al., 2016; Bell et al., 2013). In principle, we agree that this process may be occurring, but we hypothesize that it occurs only during exceptional cases of high winds and wave

high heights. The Reynolds gas transfer limitation(Zavarsky et al., 2018) occurs over a larger range of wind speeds and wave heights, but obviously does not capture all the flux limitation. Therefore, it appears that several processes may be responsible for gas transfer limitation and they are not all considered in our model. This marks the upper boundary for environmental





conditions for our model.

Table 1 shows the average offset between every data point and the linear fit ZA18. A reduction of the average offset can be seen for all data combinations. The last two columns of Table 1 show the mean absolute error. The absolute error also decreases with the application of our correction. The linear fits to the two data sets, before and after the corrections, are given in Tab. (2).

The slopes for the two corrected data sets show a good agreement. However, we do not correct for the rollover entirely. The corrected slopes are both are in the range of the linear function from SO234-2/235 <10 m s$^{-1}$ $k_{660} = 3.1 \pm 0.37 \cdot u_{10} - 5.37 \pm 2.35$(Zavarsky et al., 2018), but the slopes barely overlap within the 95 % confidence interval.

## 4.2   Nightingale parameterization

The N00(Nightingale et al., 2000) parameterization is a quadratic wind speed dependent parameterization of k. It is widely

used, especially for bulk $CO_2$ gas flux calculations as well as for DMS flux calculations in Lana11(Lana et al., 2011). The parameterization is based upon dual tracer measurements in the water performed by in the North Sea(Watson et al., 1991; Nightingale et al., 2000) as well as data from the Florida Strait (FS)(Wanninkhof et al., 1997) and Georges Bank (GB)(Wanninkhof, 1992).

We analyzed each individual measurement that was used in the parameterization to asses the amount of gas transfer limiting

instances that are within the N00 parameterization. The single measurements, which are used for fitting the quadratic function of the N00 parametrization, are shown together with N00 in the left panel of Fig. 4. As the measurement time of the dual tracer technique is on the order of days, we interpolated the wind and wave data to 1 h time steps and calculated the number of gas transfer limiting and gas transfer non-limiting instances. The right panel of Fig. 4 shows the limitation index which is the ratio of gas limiting instances to the number of data points (x-axis). The value 1 indicates that all of the interpolated one hour steps

were gas transfer limited. The y-axis of Fig. 4 depicts the relation of the individual measurement to the N00 parameterization. A ratio (y-axis) of 1 indicates that the measurement point is exactly the same as the N00 parameterization. A value of 1.1 would indicate that the value was 10 % higher than predicted by the N00 parameterization.

We expect a negative correlation between the gas transfer limitation index and the relation of the individual measurement vs the N00 parameterization. The higher the limitation index, the higher the gas transfer limitation, the lower the gas transfer

velocity k in with respect to the average parametrization. The correlation (Spearman's rank) is -0.43 with a significance level (p-value) of 0.11. This is not significant. However, we have to take a closer look at two specific points: [1] Point 11, GB11 that shows low measurement percentage despite a low limitation index, and [2] point 14, FS14 that shows high measurement percentage despite a high limitation index. GB11 at the Georges Bank showed an average significant wave height of 3.5 m, with a maximum of 6 m and wind speed between 9-13 m s$^{-1}$. As already discussed in Sect. 4.1 using the Knorr11 data set,

wave heights above 3.5 m could lead to gas transfer limitation without being captured by Reynolds gas transfer limitation model(Zavarsky et al., 2018). High waves together with the strong winds could mark an upper limit of the gas transfer limitation model(Zavarsky et al., 2018). On the other hand the FS14 data point showed an average wave height of 0.6 m and wind speed of 4.7 m s$^{-1}$. It is questionable if a flow separation and a substantial wind wave interaction can be established at this small wave height. This could mark the lower boundary for the Reynolds gas transfer limitation model(Zavarsky et al., 2018). Taking




out either or both of these measurements (GB11 or FS14) changes the correlation (Spearmans' rank) to -0.62 p=0.0233 (no GB11), -0.59 p=0.033 (no FS14) and -0.79 p=0.0025 (no GB11, no FS14). All three are significant. The black solid line in the right panel of Fig. 4 is a fit, which is based on the Eq. (14), to all points but GB11 and FS14.

$$y(x) = a_1 + a_2 \cdot \frac{1}{x - a_3} \tag{14}$$

We chose this functional form, because we follow the finding(Zavarsky et al., 2018) that the effect of gas transfer limitation is not linear but rather has a threshold. This means that the influence of limitation on gas transfer is relatively low with a small limitation ratio, but increases strongly. The fit coefficients are:$a_1 = 1.52$, $a_2 = 0.14$ and $a_3 = 1.18$ .

Figure 5 shows, according to the gas transfer model, corrected data points. A new quadratic fit was applied to the corrected data points ((Eq. 15), Fig. 5).

$$k_{660} = 0.359 \cdot u^2 \tag{15}$$

On average the new parameterization is 22 % higher than the original N00 parameterization. This increase is caused by the heavy gas transfer limitation of the individual measurements. As we believe that this limitation only affects the interfacial $k_o$ gas exchange, it might not be easily visible (decreasing k vs u relationship) in parameterizations based on dual tracer gas transfer measurements because of the potential of a large bubble influence.

### 4.3 Wanninkhof parameterization

The W14 parameterization estimates the gas transfer velocity using the natural disequilibrium between ocean and atmosphere of [14]C and the bomb [14]C inventories. The total global gas transfer over several years is estimated by the influx of the [14]C in the ocean(Naegler, 2009) and the global wind speed distribution over several years. The parameterization from W14 is for winds averaged over several hours. The WWIII model winds, used here, are 3 hourly and therefore in the proposed range(Wanninkhof,

2014). The W14 parameterization is given in Eq. (16).

$$k_{660,W14} = 0.251 \cdot (u_{10})^2 \tag{16}$$

The interesting point about this parameterization is that it already includes a global average gas transfer limiting factor. The parametrization is independent of local gas transfer limitation events. It utilizes a global, over many years averaged, gas transfer velocity of [14]C and relates it to remotely sensed wind speed. This means that the average gas transfer velocity has experienced

the average global occurrence of gas transfer limitation and therefore is incorporated in the k vs u parameterization.

The quadratic coefficient a is calculated by dividing the averaged gas transfer velocity $k_{glob}$ by $u^2$ and the wind distribution distu of u.

$$a = \frac{k_{glob}}{\sum u^2 \cdot distu} \tag{17}$$

The quadratic coefficient then defines the wind speed dependent gas transfer velocity k (Eq. (18)).

$$k = a \cdot u^2 \tag{18}$$





The left panel of Fig. 6 shows the global wind speed distribution of the year 2014 taken from the WWIII model, which is based on the NCEP reanalysis. Additionally, we added the distribution taking our wind speed correction into account. At the occurrence of gas transfer limitation we calculated, as described in Sect. 3, $u_{alt}$ as the representative wind speed for the unlimited transfer. The distribution of $u_{alt}$ shifts higher wind speed (10-17 m s$^{-1}$) to lower wind speed regimes (0-7 m s$^{-1}$). This alters the coefficient for the quadratic wind speed parametrization. A global average gas transfer velocity of $k_{glob}$=16.5 cm h$^{-1}$(Naegler, 2009) results in a coefficient a=0.2269, using the uncorrected NCEP wind speed distribution. With the $u_{alt}$ distribution a becomes 0.2439. This is an 9.85 % increase. Our uncorrected value of a=0.2269 differs from the W14 value of a=0.251 because we use a different wind speed distribution. The W14 uses a Rayleigh distribution with $\sigma = 5.83$, our NCEP derived $\sigma = 6.04$ and the corrected NCEP $\sigma = 5.78$. This means that the W14 uses a wind speed distribution with a lower global average speed. However, for correction we use the relative gas transfer reduction between our calculated parameterization and our calculated and corrected parameterization. For the calculation of a, we did not use a fitted Rayleigh function but the corrected wind speed distribution from Fig. 6.

A comparison of W14, N00 and the corrected parameterizations is shown in the right panel of Fig. 6. N00 shows the lowest relationship between u and k. W14 shows a parameterization with a global averaged gas transfer limitation influence and is therefore slightly higher than N00. It appears that the gas transfer limitation is overcompensating the smaller bubble mediated gas transfer of $CO_2$ (W14). The corrected N00 is significantly higher than the W14+9.85 %. We hypothesize that this difference is based on the different bubble mediated gas transfer of He, $SF_6$, and $CO_2$.

## 4.4 Global Analysis

We used the native global grid (0.5$^o$ x 0.5$^o$) from the WWIII for the global analysis. The data points from the DMS and $CO_2$ climatologies as well as all auxiliary variables were interpolated to this grid.

Figure 7 shows the percentage of gas transfer limited data points with respect to the total data points for every month in the year 2014. The average yearly global percentage is 18.6 %. The minimum is 15 % in March and April and the maximum is 22 % JJA. Coastal areas and marginal seas seem to be more influenced than open oceans. The reason could be that gas transfer limitation is likely to occur at fully developed seas when the wind speed is in the same direction and magnitude as the wave's phase speed. At coastal areas and marginal seas, the sea state is less influenced by swell and waves that were generated at a remote location. Landmasses block swell from the open ocean to marginal seas. The intra-annual variability of gas transfer limitation is shown in Fig. 8. Additionally, we plotted the occurrences split into ocean basins and Northern and Southern Hemisphere. Two trends are visible. There is a higher percentage of gas transfer limitation in the Northern Hemisphere and, on the time axis, the peak is in the respective (boreal and aural) summer season. The Southern Hemisphere has a water-landmass ratio of 81 %, the northern Hemisphere's ratio is 61 %. The area of free open water is therefore greater in the southern part. Fully developed seas without remote swell influence favor gas transfer limitation. In the Southern Hemisphere, the large open ocean areas, where swell can travel longer distances, provide an environment without gas transfer limitation. The peak in summer and minimum in winter can be associated with the respective sea ice extent on the Northern and Southern Hemisphere. Figure 7 shows that seas, which are usually ice covered in winter, show a high ratio of gas transfer limitation.



The global reduction of the $CO_2$ and DMS flux is shown for every month in Fig. 9 and 10. Most areas with a reduced influx of $CO_2$ into the ocean are in the northern Hemisphere. The only reduced $CO_2$ influx areas of the Southern Hemisphere are in the south Atlantic and west of Australia and New Zealand. Significantly reduced $CO_2$ efflux areas are found in the northern tropical Atlantic, especially in the boreal summer months, the northern Indian Ocean and the Southern Ocean.

For the DMS flux (Fig. 9) the absolute values of reduction, due to gas transfer limitation, coincide with the summer maximum of DMS concentration and therefore large air-sea fluxes(Lana et al., 2011; Simó and Pedrós-Alió, 1999). In the boreal winter the northern Indian Ocean also shows a high level (10 $\mu$mol m$^2$ d$^{-1}$) of reduction. The highest water concentrations and fluxes in the Indian Ocean are found in boreal summer(Lana et al., 2011), which does not seem to be greatly influenced by gas transfer limitation.

The total amount of carbon taken up by the ocean is shown in Table 3. We calculate a total carbon uptake for the year 2014 of 1.15 Pg C for the N00 parameterization without the effect of gas transfer limitation. This value is reduced by the gas transfer limitation model to 1.06 Pg C, which is a reduction of 8 %. The W14 parameterization yields an uptake of 1.16 Pg C and with the limitation model an uptake of 1.06 Pg C which is a difference of 9 %. For the parameterization used in the Takahashi climatology(Takahashi et al., 2009), we calculated a total uptake of 1.28 Pg C without gas transfer limitation. Adding the effect

of gas transfer limitation, we get a value of 1.19 Pgram C which is a reduction of 7 %. The global value from the Takahashi climatology(Takahashi et al., 2009) is 1.42 Pgram C yr$^{-1}$. Rödenbeck(Rödenbeck et al., 2015) estimate 1.75 Pg C yr$^{-1}$ as uptake between 1992 and 2009. The difference between our calculation and the estimates from the global climatologies are [1] due to the different reference year, Takahashi 2000 / Rödenbeck 1992-2009 / this study 2014, which leads to different wind speed, $\Delta pCO_2$ and SST data. [2] The data set and influence for sea ice cover is different. However, the estimated reduction of

7-9 %, due to gas transfer limitation, is also valid for the Takahashi and Rödenbeck estimates.

The DMS emissions from the ocean to the atmosphere are shown in Table 4. The calculated total emission from the N00 parameterization is 50.72 Tg DMS yr$^{-1}$ for the year 2014. This is reduced, due to our gas transfer limitation calculations, to 45.47 Tg DMS yr$^{-1}$, which is a reduction of 11 %. The linear parameterization ZA18 estimates an emission of 56.22 Tg DMS yr$^{-1}$. Using the gas transfer limitation model the linear parameterization is reduced to 51.07 Tg DMS yr$^{-1}$,

which is a reduction of 11 %. Global estimates are 54.39 Tg DMS yr$^{-1}$(Lana11(Lana et al., 2011)) and 45.5 Tg DMS yr$^{-1}$(Lennartz15(Lennartz et al., 2015)). Similar to the reasons we mentioned in the paragraph above, a difference in wind speed or sea ice coverage could be the reason for the difference in the global emission estimated between the Lana climatology and our calculations with the N00 parameterization. Lennartz15(Lennartz et al., 2015) uses the water concentrations from the Lana climatology, but includes air-side DMS concentrations, which reduces the flux by 17 %. We do not include air-side DMS concentrations but gas transfer

limitation, which reduces the flux by 11 %. Including both processes we can expect a reduction of 20-30 %.

The global $CO_2$ air-sea flux is reduced by 7-9 % due to gas transfer limitation. The impact on the DMS climatology is 11 %. This is in the range of 9.85 % which is the estimated influence of gas transfer limitation on the W14 parametrization through a different wind speed distribution. The different reduction percentages between these two gases are attributed to the larger bubble mediated gas transfer of $CO_2$, which compensated the loss of flux for $CO_2$ but not for DMS.



## 5   Conclusions

We provide a model to correct for the gas transfer limitation due to wind-wave interaction(Zavarsky et al., 2018). $Re_{tr}$ and the resulting alternative wind speed $u_{alt}$ can be calculated from standard meteorological and oceanographic variables. Additionally the condition (period, height, direction) of the ocean waves have to be known or retrieved from wave models. The calculation is

iterative and can be easily implemented. The effect of the correction is shown with two data sets from the Knorr11(Bell et al., 2017) and the SO234-2/235 cruise (Zavarsky et al., 2018). Both data sets show, after the correction, a better agreement with the linear ZA18 parameterizations (Table 1and Table 2), which only contains non limited gas transfer velocity measurements from the SO 234-2/235 cruise. Generally, the correction may be only applied to the interfacial gas transfer velocity $k_o$.

We investigated the individual measurements leading to the N00 gas transfer parameterization for the influence of gas transfer

limitation. We think that the overall parameterization is heavily influenced by gas transfer limitation but, due to the measurement method (dual tracer measurements), the limitation is masked by bubble mediated gas transfer. We show a significant negative correlation between the occurrence of gas transfer limitation and the ratio of the individual measurement to the N00 parameterization. We applied a gas transfer limitation correction and fitted a new quadratic function to the corrected data set. The new parameterization is on average 22 % higher than the original N00 parameterization. This leads to the conclusion that

gas transfer limitation influences gas transfer parameterizations, even if it is not directly visible, via a smaller slope.

For the W14 parameterization we used a global wind speed climatology for the year 2014 and applied the gas transfer limitation model $u_{10} \rightarrow u_{alt}$. Using the distribution function of $u_{alt}$ we calculated a corrected gas transfer parameterization. The coefficient of the corrected parameterization is 9.85 % higher than the original one. W14 already includes the global average of gas transfer limitation. Therefore the increase, due to the correction, is expected to be less than the one for N00. The uncorrected

N00 is lower than W14, but after correction N00 is larger than the corrected W14, which is expected due to the larger bubble mediated gas transfer of He and $SF_6$ over $CO_2$.

In addition, we calculated the global carbon uptake of $CO_2$ due to air-sea exchange and the global emission of DMS. The reduction, due to the consideration of gas transfer limitation, is between 7-9 % for $CO_2$ and 11 % for DMS. This is in the range of the calculated influence of gas transfer limitations on the global parameterization W14.

We think that gas transfer limitation has a global influence on air-sea gas exchange of 7-11 %. These numbers are supported by the correction of the W14 parametrization as well a global DMS and $CO_2$ gas transfer calculation. Local conditions may lead to much higher influences. Gas transfer velocity parameterizations from regional data sets might be heavily influenced by gas transfer limitations. We have shown this for the N00 parameterization. This should be considered with their use.

For global calculations we recommend the use of the Wanninkhof parameterizations(Wanninkhof, 2014), as it already has an

average global gas transfer limitation included. We recommend using a linear parameterization (e.g. ZAV17) for rather soluble gases, such as DMS, in the cases of non-limited gas transfer. The limitation can be determined using the $Re_{tr}$ parameter. If conditions favor limitation, we recommend our iterative approach to correct u to $u_{alt}$ (Fig. 1). For gases with a similar solubility as $CO_2$, we recommend the use of W14. In case of no gas transfer limitation, we recommend the used of the corrected





W14+9.85 % parameterization. The corrected N00 (N00+22 %) parameterization is recommended for very insoluble gases with the absence of gas transfer limitation, the original N00 is recommended for the gas transfer limited case.

*Data availability.* The wave data is available at the website of the NOAA Environmental Modelling Center. The ERA-Interim data is available at the website of the ECMWF. The data is stored at the data portal of GEOMAR Kiel.

5  *Competing interests.* The authors declare no competing interests.

*Acknowledgements.* The authors thank Kirstin Krüger, the chief scientist of the RV Sonne cruise (SO234-2/235), as well as the captain and crew. We thank the Environmental Modeling Center at the NOAA/National Weather Service for providing the WaveWatch III data. We thank the European Centre for Medium-Range Weather Forecasts for providing the ERA-Interim data. This work was carried out under the Helmholtz Young Investigator Group of C. Marandino, TRASE-EC (VH-NG-819), from the Helmholtz Association. The cruise 234-2/235
10  was financed by the BMBF, 03G0235A.



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





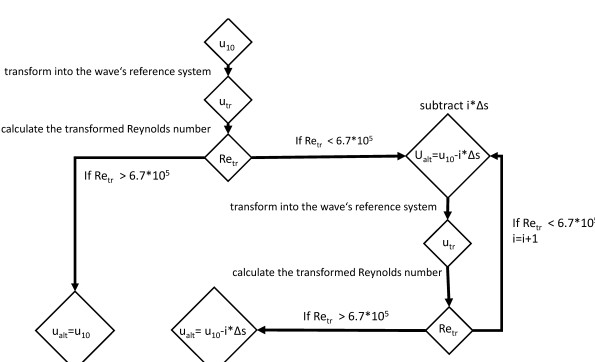

**Figure 1.** Work flow of the gas transfer limitation model. In the case of limited gas transfer, the output is the corrected wind speed $u_{alt}$, which then can be used in gas transfer parameterizations. The step size $\Delta s$ can be adapted freely, but considerations of resolution and computing power have to be made. For this manuscript we set $\Delta s = 0.3 \ m \ s^{-1}$.




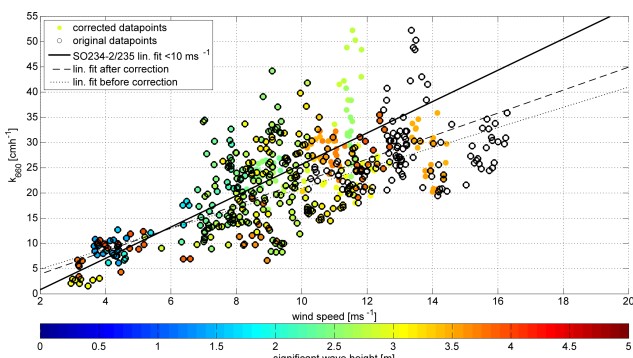

**Figure 2.** Correction of the SO234-2/235 DMS fluxes. The data points with $\mathrm{Re}_{tr} < 6.7 \cdot 10^5$ were corrected using the gas transfer limitation model. Black circles denote k values at the original wind speed $u_{10}$. Colored filled circles denote the k value at wind speed=$u_{alt}$. The color shows the significant wave height. If a data point has a concentric black and filled circle, it was not corrected as it was not subject to gas transfer limitation. The black solid line is the ZAV17 parameterization. The dotted line is the linear fit to the data points before the correction, the dashed line is the linear fit after the correction.





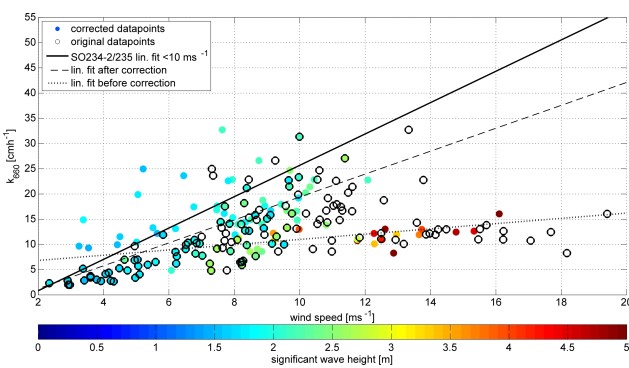

**Figure 3.** Correction of the Knorr11 DMS fluxes. The data points with $\mathrm{Re}_{tr} < 6.7 \cdot 10^5$ were corrected using the gas transfer limitation model. Black circles denote k values at the original wind speed $u_{10}$. Colored filled circles denote the k value at wind speed=$u_{alt}$. The color shows the significant wave height. If a data point has a concentric black and filled circle, it was not corrected as it was not subject to gas transfer limitation. The black solid line is the ZAV17 parameterization. The dotted line is the linear fit to the data points before the correction, the dashed line is the linear fit after the correction.





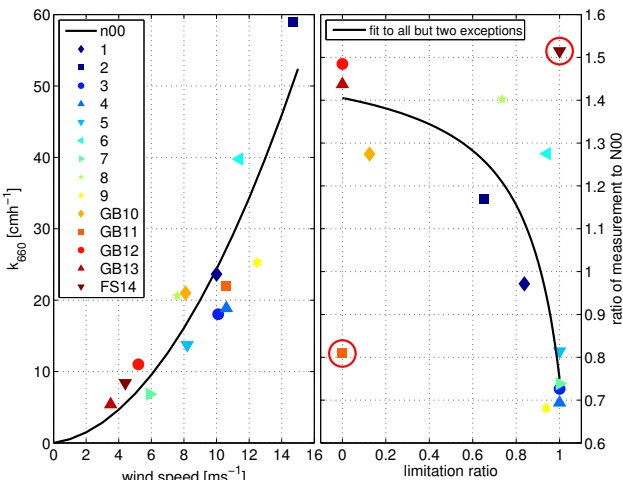

**Figure 4.** Individual dual tracer measurements which contribute to the N00 (solid line) parameterization [left panel]. The relationship of the gas limitation ratio to the measurement/N00 ratio [right panel]. A higher limitation ratio indicates a longer influence of gas transfer limitation on the data point. The solid line in the right panel is a fit to the limitation to measurement/N00 relationship. The two red circles denote the outlier points which are discussed in the text. The black solid line is a fit using the function $y(x) = a_1 + a_2 \cdot \frac{1}{x - a_3}$. The fit coefficients are: $a_1 = 1.52$, $a_2 = 0.14$ and $a_3 = 1.18$ .




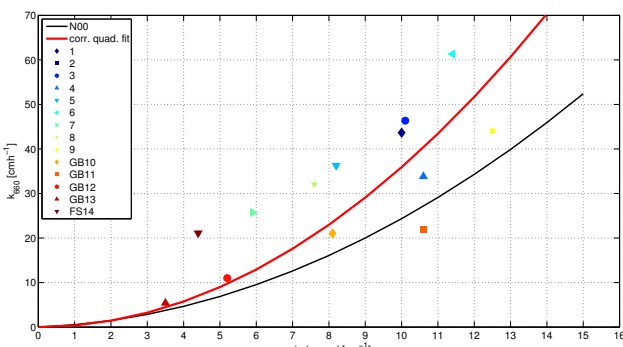

**Figure 5.** Corrected individual measurements, comprising the N00 parameterization, resulting from the algorithm described in Sect. 3. The difference between $u_{alt}$ and the original $u_{10}$ was added to k using the linear parameterization ZAV17. This is correcting the limitation of $k_o$ due to wind-wave interaction. The black solid line is the original N00 parametrization. The red line is a new quadratic fit to the corrected data points $k=0.359*u^2$.





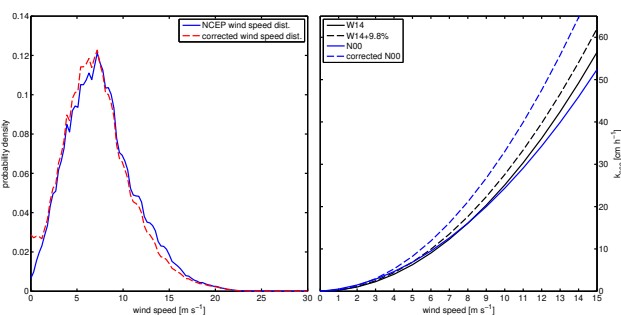

**Figure 6.** Wind speed distributions for the year 2014 [left panel]. The solid line is NCEP derived wind speed distribution, the dashed line the wind speed distribution of the corrected wind speed $u_{alt}$. Comparison of original and limitation corrected k vs wind speed parameterizations [right panel].



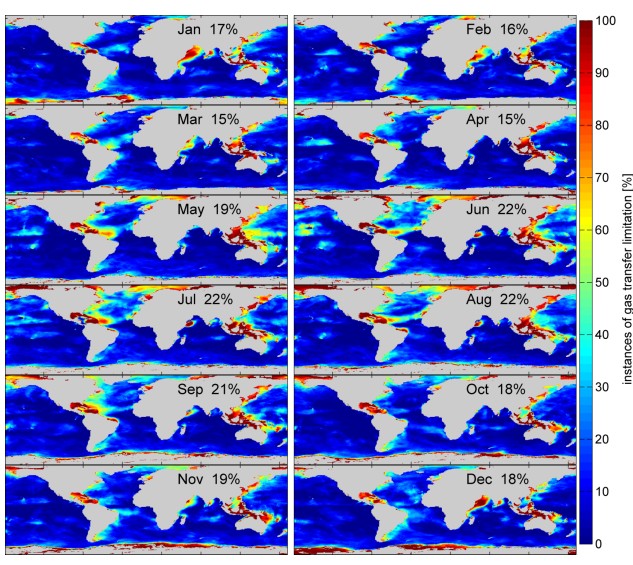

**Figure 7.** The global probability of experiencing gas transfer limitation during the respective month (2014). The percentage is the number of gas transfer limited occurrences with respect to the total data points with a 3 h resolution.





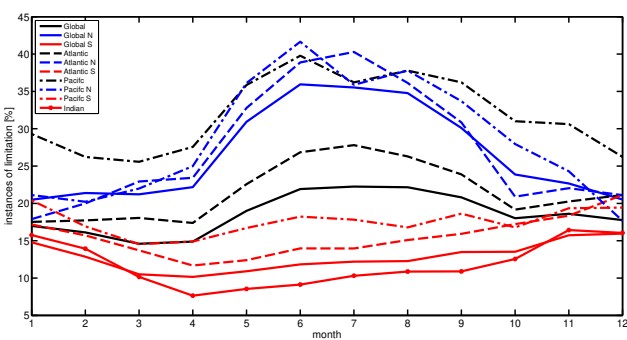

**Figure 8.** The probability of experiencing gas transfer limitation during the respective month (2014) divided into ocean basins and hemisphere. The Southern Ocean was added to the southern part of the respective ocean basin. The percentage is the number of gas transfer limited instances with respect to the total data points with a 3 h resolution.





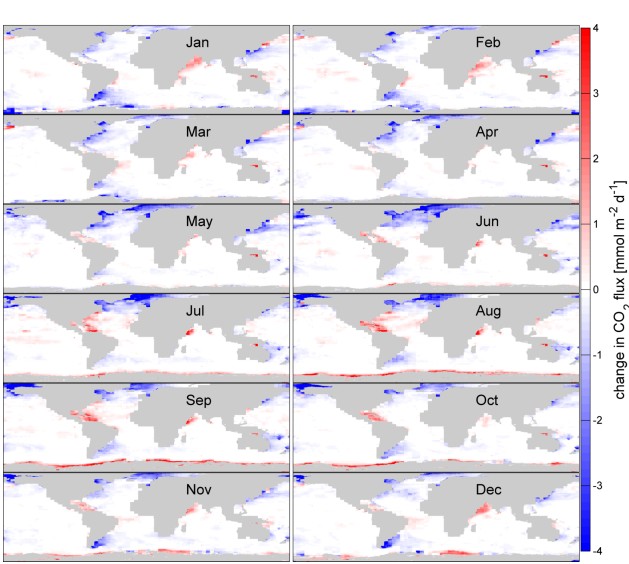

**Figure 9.** The absolute change of $CO_2$ gas transfer due to limitation for each month of 2014. Negative values (blue) denote areas where a flux into the ocean is reduced by the shown value. Positive values denote areas where flux out of the ocean is reduced by the shown value.





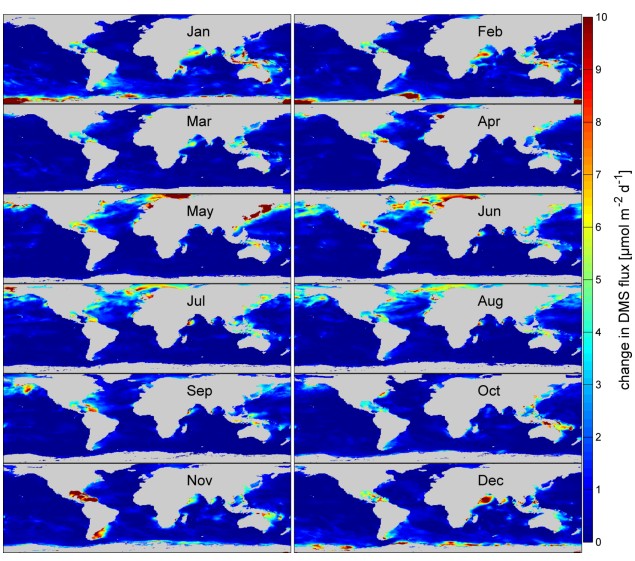

**Figure 10.** The absolute change of DMS gas transfer due to limitation for each month of 2014. The shown magnitudes denote the reduction by gas transfer limitation.



| reference fit<br>all [cm h$^{-1}$] | SO234-2/235<br>mean diff. | Knorr11<br>mean diff. | SO234-2/235<br>mean(‖) | Knorr11<br>mean(‖) |
|---|---|---|---|---|
| lin. fit SO234-2/235 to corrected | -1.2 | -6.7 | 5.5 | 8.1 |
| lin. fit SO234-2/235 to uncorrected | -2.8 | -10.3 | 6.4 | 10.7 |

**Table 1.** Mean differences between the fits in column one and the corrected and the uncorrected k data sets. A negative value describes that the fit, on average, overestimates the actual measured data. The mean of the absolute value is presented in the last two columns.





|  | Knorr11 | SO234-2/235 |
|---|---|---|
| uncorrected | $k_{660} = 0.52 \pm 0.4 \cdot u + 5.79 \pm 4.82$ | $k_{660} = 2 \pm 0.42 \cdot u + 0.94 \pm 2.48$ |
| corrected | $k_{660} = 2.27 \pm 0.5 \cdot u - 3.29 \pm 4.08$ | $k_{660} = 2.28 \pm 0.45 \cdot u - 0.63 \pm 4.14$ |

**Table 2.** Linear fits to the corrected and uncorrected data sets of Knorr11 and SO234-2/235. The error estimates correspond to a 95 % confidence interval.



| parameterization | flux [Pg C] |
|---|---|
| N00 | 1.15 |
| N00 $Re_{tr}$ | 1.06 |
| W14 | 1.16 |
| W14 $Re_{tr}$ | 1.06 |
| Tak09 | 1.28 |
| Tak09 $Re_{tr}$ | 1.19 |
| Takahashi 2009(Takahashi et al., 2009) | 1.42 Pg yr$^{-1}$ |
| Rödenbeck(Rödenbeck et al., 2015) | 1.75 Pg yr$^{-1}$ |

**Table 3.** 2014 carbon flux in Pg. $Re_{tr}$ indicates an application of the gas transfer limitation model. The last two lines are estimates from previous published work.



| parameterization | flux [Tg DMS yr$^{-1}$] |
|---|---|
| N00 | 50.72 |
| N00 Re$_{tr}$ | 45.47 |
| ZA18 | 56.22 |
| ZA18 Re$_{tr}$ | 51.07 |
| Lana11(Lana et al., 2011) | 54.39 Tg DMS yr$^{-1}$ |
| Lennartz15(Lennartz et al., 2015) | 45.5 Tg DMS yr$^{-1}$ |

**Table 4.** 2014 DMS flux in Tg. Re$_{tr}$ indicates an application of the gas transfer limitation model. The last two lines are estimated from global climatologies