# Peer review of "The influence of transformed Reynolds number suppression on gas transfer parameterizations and global DMS and CO2 fluxes"

_Atmospheric Chemistry and Physics, 2018_

## Referee Comment (RC1) · C. Fairall (Referee) · 18 May 2018

This paper describes modifications to water-side viscous transfer velocity, k0, parameterizations to account for reductions in transfer associated with hypothesized flow separation that is diagnosed via a wave-based scaling parameter, Retr. Retr is defined as a Reynolds number in a reference frame moving with the wave peak phase speed. If the abs(Retr)<6.7e5, then k0 is suppressed ('limited') because of flow separation on the downwind side of the peak waves. The physical mechanism is that flow separation reduces the area of strong viscous coupling on the air side and this limits the forcing

of the molecular sublayer transfer on the ocean side. The details are given in a previous paper – ZA18. The authors use these results to recalibrate a number of different parameterizations to account for the transfer suppression. For example, for DMS the transfer coefficient is assumed linear an represented as $k0=a*u-b$ where $a=3.1$ and $b=-5$, $u=u10$ if $ABS(Retr)>6.7E5$ and $u=ualt$ if $ABS(Retr)<6.7E5$. The modified wind speed, ualt, is computed by reducing u10 until $ABS(Retr)=-6.7E5$. The authors showed that remapping the wind speed in this manner leads to a more linear k vs u10 behavior from their two data sets. They also apply a similar remapping to the Nightingale 2000 data set so that $k=c*u^2$ where $c=0.36$, which is 22% higher than the unmapped fit. Note k here seems to include but viscous and bubble-mediated transfer. Finally they use this approach to compute global fluxes of DMS and CO2 with various k representations and find that their new approach reduces fluxes about 10%.

In my view a reduction on global fluxes of 10% is significant enough to justify publication. However, the paper appears to be hastily written and not carefully crafted to make it easy for the reader. It is hard to read, difficult to follow and contains a bewildering variety of coefficients, percentages, and information that is poorly organized. I have no confidence I actually know which coefficients were used where. On a side note, my own opinion is that the phenomenon they are characterizing (reduction in k in certain air-sea conditions) is almost certainly not flow separation but it is possible their use of Retr is capturing a lot of what is happening. Since ZA18 is published, I think my skepticism should not prevent publication of this paper and I don't want to argue that point here.

Here are some specific comments:

P2 line 18 Suggest identifying ZA18 here and using it throughout.

P2 Line 21 In ZA18 theta is defined as the angle between the wave direction and utr, which would be at 90 deg to what is said here. I am somewhat confused by fig D1 in ZA18 which states both that the 'wave crests are moving to the left' and that 'The wave

travels from left to right'.

P3 line 14 'W14 must already have gas transfer limitation included as it is solely dependent on carbon isotopes to estimate the air-sea flux over several years'. Not sure what you mean here. Are carbon isotopes relevant to this? Do you mean that the mean flux is associated with averages that include the mean contribution of non-limited and limited conditions? I think you mean it is applicable to average wind speed conditions as is. Please clarify.

P4 line 25. You might not want to push the 'flow separation' aspect since it is not necessary to your development here.

P5 line As I interpret the mathematics, reducing the wind speed until ABS(Retr)=6.7E-5 will cause Retr=-6.7E-5. Also, I don't see why you need to iterate. Is it not true that

ualt=(cp-6.7E5*nu/Hs)/cos(th1-th2)

where th1 is the direction of the wind and th2 is the direction of the waves in the earth frame?

P5 line 13 It looks like you change the wind speed because that is in the k parameterization.

P5 line 29 'We subtract a linear dependency using the ZA18 parameterization, to account for the gas transfer limitation in k0'. I don't understand this. I think you need to provide a few equations to make it clear. Eq (13) looks wrong to me; you have removed most of the linear part and left only the bubble part. Should it be k*(u10-ualt) instead? Also, do you actually use (13) anywhere? I do not see it referred to anywhere else in the document. Is NI00 the same as N00?

P6 lines 6-7 You said that already.

P6 line 15 Change to 'plotted at the corrected..'

P6 line 17. Is ZAV17 the same as ZA18?

P6 line 23 Add 'for Knorr11' after 'estimates'.

P6 line 26 The explanation about waves greater than 3.5 m/s seems to be total speculation. It is true that large swell are unlikely to have flow separation.

P6 line 31-32 Perhaps reiterate this in the conclusions.

P6 line 9 Suggest giving the formula for N00 as you did for W14.

P7 line 8 Just to be clear, k corrected = k*y(x), yes?

P8 line 23 Not sure the parameterization is 'independent' of events. I think you mean it includes them in a globally average sense. Your Fig. 8 implies that it would vary regionally.

P9 line 16. Since N00 applies to less soluble gases with more bubble enhancement, it seems like the limitation should be less. You argue it is 'masked' by bubbles.

P9 Section 4.4 Please state exactly and clearly formulas used in the computations. Suggest adding the formulas to Tables 3 and 4 with a clear rendition of their application.

---

## Author Comment (AC1) · 12 Jul 2018

Fairall:"In my view a reduction on global fluxes of 10% is significant enough to justify publication. However, the paper appears to be hastily written and not carefully crafted to make it easy for the reader. It is hard to read, difficult to follow and contains a bewildering variety of coefficients, percentages, and information that is poorly organized. I have no confidence I actually know which coefficients were used where. On a side note, my own opinion is that the phenomenon they are characterizing (reduction in k in certain air-sea conditions) is almost certainly not flow separation but it is possible their

use of Retr is capturing a lot of what is happening. Since ZA18 is published, I think my skepticism should not prevent publication of this paper and I don't want to argue that point here."

We thank Dr. Fairall for his review and helpful comments. We would be interested in hearing, perhaps off-line, why he does not think that flow separation is a likely suspect. Regarding the manuscript writing, we have gone through the text thoroughly to try to minimize confusion. We hope that we have adequately modified the manuscript with regard to both clarity and to the points raised below.

Fairall:"Here are some specific comments: P2 line 18 Suggest identifying ZA18 here and using it throughout."

We aim to use ZA18 just like we use N00 or W14, namely to refer to the respective k vs u parameterization and not to the published paper. Therefore, we would like to keep the full reference as a citation and when the parameterization is used we use ZA18.

Fairall:"P2 Line 21 In ZA18 theta is defined as the angle between the wave direction and utr, which would be at 90 deg to what is said here."

We changed the wording here, especially as our use of wave crest is unclear.

Retr is the Reynolds number transformed into the reference system of the moving wave. utr is the wind speed transformed into the wave's reference system, Hs the significant wave height, $\eta$air the kinematic viscosity of air and $\Theta$ the angle between the wave direction and direction of utr

Fairall:"I am somewhat confused by fig D1 in ZA18 which states both that the 'wave crests are moving to the left' and that 'The wave C2 travels from left to right'."

You are right. It is a mistake in the subtext of the figure. This has been reported to the production editors and should be corrected in the final version. The figure is the correct one.

[Figure]

Fairall:"P3 line 14 'W14 must already have gas transfer limitation included as it is solely dependent on carbon isotopes to estimate the air-sea flux over several years'. Not sure what you mean here. Are carbon isotopes relevant to this? Do you mean that the mean flux is associated with averages that include the mean contribution of non-limited and limited conditions? I think you mean it is applicable to average wind speed conditions as is. Please clarify."

We changed the wording in this paragraph to make it clearer. A few more words to make our point: Since W14 is based on the amount of 14C in the global oceans, the gas transfer from source region (atmosphere) to sink region (ocean) encompasses all the processes we seek to understand by investigating air-sea gas exchange. Therefore, the gas transfer suppression mechanism is also included. Other measurement techniques, such as dual-tracer or eddy covariance, act on different scales: Minutes to days and a few kilometers on the spatial scale. These techniques, because of their local and short spatial and temporal scales, might not experience all processes influencing gas transfer at every measurement. This might be an advantage to identify new processes, but has the major disadvantage that gas transfer parameterizations based on one or the other dataset might not include certain important processes, such as gas transfer suppression.

Fairall:"P4 line 25. You might not want to push the 'flow separation' aspect since it is not necessary to your development here."

We are describing the process that causes, in our opinion, gas transfer suppression. The model reduces the wind speed u10 to a windspeed ualt, which according to the transformed Reynolds number, does not create flow separation and therefore no gas transfer suppression. We would like to keep our hypothesis in this text.

Fairall:"P5 line As I interpret the mathematics, reducing the wind speed until ABS(Retr)=6.7E-5 will cause Retr=-6.7E-5. Also, I don't see why you need to iterate. Is it not true that ualt=(cp-6.7E5*nu/Hs)/cos(th1-th2) where th1 is the direction of

the wind and th2 is the direction of the waves in the earth frame?"

The wind speed in the earth's reference system must be transformed into the wave's reference system by subtracting the respective vectors. Then the angle between the wind direction and the wave moving in the v direction (for example from east to west) has to be calculated. You describe the relative directions in the earth's reference system. For our model, however, everything must be transferred into the wave's reference system. The transformation is described in Appendix C of Zavarsky et al., (2018). We do not think ualt is analytically solvable, because th1 and ualt change in the wave's reference system with a change of u10, which then changes in the earth's reference system.

Fairall:"P5 line 13 It looks like you change the wind speed because that is in the k parameterization."

Of course, the most widely used gas transfer parameterizations are related to wind speed. We think that gas transfer suppression is a wind-wave interaction. We change the wind speed because it is one of the major factors in gas transfer and flow separation. The wind speed (in the wave's reference system) is obviously influencing the flow separation. We set the wave field as constant or externally prescribed. Therefore, we use the wind speed as parameter. A change in wind speed u10 (earth's reference system) changes utr in the wave's reference system. If we change by iteration u10 and as a consequence utr we can evaluate at which u10 flow separation sets in and can use this to correct for it.

Fairall:"P5 line 29 'We subtract a linear dependency using the ZA18 parameterization, to account for the gas transfer limitation in k0'. I don't understand this."

For the ZA18 parameterization u10 is just replaced by ualt . ZA18 is a parameterization based on DMS gas transfer measurements. DMS gas transfer velocities are not greatly influenced by bubble mediated gas transfer. We think that gas transfer suppression only affects interfacial gas transfer. Therefore, we subtract a parameterization

which reflects interfacial gas transfer only from the non-linear relationships that include bubble-mediated effects. Please see the answer to the next comment as well.

Fairall:"I think you need to provide a few equations to make it clear. Eq (13) looks wrong to me; you have removed most of the linear part and left only the bubble part. Should it be k*(u10-ualt) instead? Also, do you actually use (13) anywhere? I do not see it referred to anywhere else in the document. Is NI00 the same as N00?"

You are right about equation 13, but it should be k*(u10)-k*ualt. I corrected that. It was correct in the Matlab code, so the data is still right. I also added another equation to make it clear. Sorry, we used two abbreviations for the Nightingale 2000 parameterization. NI00 was replaced throughout by N00.

Fairall:"P6 lines 6-7 You said that already."

We deleted these two sentences. Thanks.

Fairall:"P6 line 15 Change to 'plotted at the corrected...'"

We changed it accordingly. Thanks.

Fairall:"P6 line 17. Is ZAV17 the same as ZA18?"

We removed ZA17. Capital letters and the year number do not refer to the paper, but to the parameterization derived from this paper.

Attached is also an updated version of the manuscript.

Please also note the supplement to this comment:
https://www.atmos-chem-phys-discuss.net/acp-2018-32/acp-2018-32-AC1-supplement.pdf

**Supplement:**

[revised manuscript text omitted]
 suppression and is a representation of $k_o$ as DMS does not have a significant contribution of bubble mediation gas transfer. For ZA18 $u_{alt}$ can be directly inserted into this parameterization as we do not expect a large bubble contribution to k (Eq. (12)). However, all other parameterizations are based on measurements with rather insoluble gases, which have a significant bubble mediated gas transfer contribution. As a consequence we subtract a linear dependency using the ZA18 parametrization, to account for the gas transfer suppression in

$k_o$ (Eq. (13 and14 )).

$$F_{lim,ZA18} = k_{ZA18}\left(u_{alt}\right) \cdot \Delta C = \left(3.1 \cdot u_{alt} - 5.37\right) \cdot \Delta C \tag{12}$$

$$F_{lim,Tak00/W14/N00} = \left[k_{Tak00/W14/N00}\left(u_{10}\right) - \left(k_{ZA18}\left(u_{10}\right) - k_{ZA18}\left(u_{alt}\right)\right)\right] \cdot \Delta C \tag{13}$$

$$F_{lim,Tak00/W14/N00} = \left[k_{Tak00/W14/N00}\left(u_{10}\right) - 3.1 \cdot \left(u_{10} - u_{alt}\right)\right] \
[revised manuscript text omitted]

---

## Referee Comment (RC2) · M. Yang (Referee) · 6 Sep 2018

This paper looks at the implications of air flow separation on previously published gas transfer velocities as well as on the global oceanic CO2 and DMS fluxes. The paper relies heavily on the recently published ZA18 from the same group, which argues that when wind and waves are aligned the leeside of the wave is sheltered from the wind and encounters less turbulence. This theory was used by ZA18 to explain the fairly low transfer velocities of DMS and CO2 during a recent Indian Ocean cruise at wind speeds over 10 m/s. The first reviewer has already given a detailed review, pointing out

some mathematical inaccuracies. The authors have supplied a new, revised version of the manuscript. It is this revised version that I will comment on.

Overall, I find this paper still very confusing and the main results self-contradictory. The abstract states that corrections of Nightingale et al 2000 and Wanninkholf 2014 for air flow separation leads to INCREASES in the gas transfer velocity. However, applications of these corrected k parameterizations led to ~10% DECREASES in the global flux magnitudes. This doesn't make any sense from a superficial level. Figures 2 and 3 show that correcting for air flow separation moves wind speed to the left (i.e. Ualt <U10 when there's suppression), while k remains unchanged. Figure 6 (left panel) shows that correcting for air flow separation moves the global wind speed distribution to the left. Is it this adjustment in wind (rather than an adjustment to k) that causes the global fluxes to reduce in magnitude? But then Figure 4 and 5 show that the actual k values (the individual dual tracer points) are adjusted upwards (rather than the wind speeds adjusted leftwards) for the Nightingale et al 2000 data. How? Not clear. Seems to me that there are two possible philosophical approaches: a) U10 is a good predictor of k, and thus points below the expected mean k vs. U10 relationship (suppressed data) should be adjusted upwards in terms of k, or b) U10 is not a good predictor of k as waves are also important; thus we either need a new x variable that includes wave-wind interaction, or adjust U10 to account for waves as the authors have done in the Ualt calculation. But in this paper the authors seem to be taking both of these approaches.

The bottom line is that the authors are trying to address an interesting and important topic. Unfortunately I find the paper far from publishable currently. And so I recommend a major revision and give the authors a chance to clearly address the issues raised.

Comments:

At a given wind speed, there is probably a range of gas transfer velocities as a function of sea state. Recent works from Blomquist et al 2017 and Brumer et al 2017

demonstrate the efficacy of the 'wave-wind Reynolds number'. Some of the variability in sea state may be encapsulated by the authors' transformed Reynolds number, fine. At expected times of gas transfer suppression, the authors decided to adjust the wind speed (U10) downwards to a transformed wind speed (Ualt) by using a threshold in the transformed Reynolds number. I think this binary treatment (i.e. either suppressed or not suppressed, instead of varying degrees of sea state effect) is overly simplistic. What is the quantitative reasoning for adjusting wind speed downwards to the threshold REtr value in the case of suppression? Why not adjusting to an even lower |REtr| value, for example? And could there be times when k is 'enhanced' relative to the mean relationship?

For DMS (Figures 2 and 3), k is simply shifted to the left due to the U10 to Ualt correction. However, Figure 5 shows that the dual tracer k values from Nightingale et al 2000 are actually shifted upwards, while wind speed remains unchanged. It looks like R2 is worse in Figure 5 than in Figure 4. How did the authors make this latter correction (Eq. 14?) and why the inconsistency in approach? The authors did not apply the U10 to Ualt correction to the N00 data, as with DMS, because N00 data are more affected by bubbles? Also, the authors implied that the N00 dataset were taken in places (many coastal) and during times when gas transfer suppression is predicted to happen more often than the global average. Following that logic, shouldn't the global fluxes be higher, and not lower, if the original N00 contained a lot of suppressed gas transfer data?

In the case of W14, it is a single global average point averaged over multiple years. It presumably does include the full range of sea states. This single k point is pinned against a global mean wind speed (accounting for wind distribution). So if the functionality of W14 is correct, I don't see how it needs to be corrected at all to account for air flow separation. Does the right panel of Fig 6 imply an upward adjustment in the k value, or a leftward adjustment in wind speed? Shouldn't fluxes computed from [original W14 x NCEP wind speed distribution] be the same as those computed from

[adjusted W14 x corrected wind speed distribution]? It's worth noting that in the revised wind distribution, there is far more occurrence of 'zero wind speed', which in the W14 formulation would result in zero flux. Are the authors saying that under conditions of moderate-to-high wind speed, when wind and waves follow each other, there is no gas transfer?

Some technical comments: Your Eq. 1 is presented from the perspective of air concentrations. Since you're talking about water-side controlled gases, it seems more appropriate to present this Eq. from the perspective of water concentrations, i.e. k * (Ca*H - Cw) or k * (Ca/H - Cw), depending on whether your H is water to air or air to water. Also, your Eq. 1 adopts the convention of positive flux into the ocean. That's consistent in sign to your global CO2 flux, but not to your DMS flux. Please be consistent.

Eq. 3: I think you have left out the H term. Should be 1/ktot = 1/kw + H/ka (if H is water to air)

In many of the plots, I think it's misleading to call Ualt 'wind speed' on the x-axis, and have both k vs U10 and k vs Ualt on the same plot.

Figures 9 and 10. Which original parameterization is used? Please specify in the captions.

Finally, the results from ZA18 are heavily used in this current paper. While the presented argument of air flow separation is a neat theory, I don't think it's well backed up by the observations for at least three reasons: 1) It is conceivable that transfer velocity varies with the directional difference between wind and wave as well as with the relative wind velocity relative to the wave phase speed. However, I don't understand the directional dependence in the formulation of the transformed Reynolds number. Air flow separation and sheltering are argued to occur when wind and waves are aligned (and not occur when they are orthogonal). However, cos (0) = 1 and cos (90) = 0. And it is a low transformed Reynolds number that is argued to cause a suppression (or limitation) in gas transfer. This seems contradictory. 2) in ZA18, the authors attempted to

explain previous transfer velocity datasets with the flow separation theory, but did not use actual (in situ or modeled) wave data. This is a significant shortcoming in my view. I have the ECMWF wave and in situ wind data from those cruises. It is not obvious that waves from cruises when gas transfer suppression were observed differed obviously from the waves during other cruises. The authors are welcomed to contact me and use these data to further improve their work. 3) a lot of high points and noise in the kDMS and kCO2 data occurred when the delta C were very small. Not only are fluxes very noisy under these conditions, any small bias in delta C would also significantly affect the derived k. Is there still a noticeable 'suppression' if the authors remove these low delta C points?

These last comments are not criticisms of the ACPD paper, but partly explain why I find the current paper rather unconvincing.

---

## Author Response (AR1)

M. Yang (Referee) miya@pml.ac.uk

**This paper looks at the implications of air flow separation on previously published gas transfer velocities as well as on the global oceanic CO2 and DMS fluxes. The paper relies heavily on the recently published ZA18 from the same group, which argues that when wind and waves are aligned the leeside of the wave is sheltered from the wind and encounters less turbulence. This theory was used by ZA18 to explain the fairly low transfer velocities of DMS and CO2 during a recent Indian Ocean cruise at wind speeds over 10 m/s.**

**The first reviewer has already given a detailed review, pointing out some mathematical inaccuracies. The authors have supplied a new, revised version of the manuscript. It is this revised version that I will comment on. Overall, I find this paper still very confusing and the main results self-contradictory. The abstract states that corrections of Nightingale et al 2000 and Wanninkholf 2014 for air flow separation leads to INCREASES in the gas transfer velocity. However, applications of these corrected k parameterizations led to ∼ 10% DECREASES in the global flux magnitudes. This doesn't make any sense from a superficial level.**

We thank Dr. Yang for his constructive comments on our paper and we think that the manuscript is much clearer now as a result. In an attempt to make the manuscript less confusing, we highlight the goals of the paper here and in the revised version. The revised version follows directly after this response. All changes are marked yellow.  The first goal is to show that gas transfer suppression happens frequently in situ and appears in our most used gas transfer velocity parameterizations. The second goal is to provide other scientists, who compute fluxes but do not measure them directly, with a simple way to account for the suppression during their studies (i.e. a u correction). With regard to Dr. Yang's comment about our self-contradictory results, we hope that by making our text clearer we have adequately shown that our results are consistent. Below are some arguments in support of our stated goals (underlined text is sometimes used to directly answer Dr. Yang's comments):

Goal 1) If gas transfer suppression is ubiquitous, scientists who compute fluxes using wind speed based parameterizations and in situ measurements of concentration difference will need to correct for it. We wanted to provide a rather simple approach for them to use and wind speed seemed to be the key. It is always measured and they are already used to using this term in their calculations. They would need to compute their transformed Reynolds number (which includes some easy vector calculations), but once that is done they can easily iterate their wind speed to compute a lower and more suitable k for their presumed suppression. This is much more doable than using a different k parameterization based on physical quantities that are not typically measured.

Goal 2) We thought it is strange that the Nightingale 2000 parameterization has, although based upon $^3$He/SF$_6$ dual tracer measurements, a flatter k vs u slope than the Wanninkhof 2014 parameterization or the k values determined by CO$_2$ eddy covariance (Wanninkhoff and Mc Gillis 1999). Dual tracer based gas transfer parameterizations should exhibit a higher bubble mediated gas transfer contribution than those based on CO$_2$ measurements, due to the lesser solubility of $^3$He and SF6. Therefore, we think that a suppressed interfacial gas transfer reduces the total gas transfer during the studies used for the Nightingale 2000 parameterization, although this decrease is not visible. Soloviev 2007, in figure 5(b), shows a case when interfacial gas transfer reduction is superimposed by the bubble term (the slope is subsequently influenced, but an overall decrease with u is not seen).

We think the reason why Nightingale 2000 can be used for CO$_2$ gas transfer calculations (the data in Zavarsky 2018a also follows N00) is that the data set is under the influence of gas transfer suppression, which balances the high bubble mediated transfer of the dual tracer data set. We thought it would be interesting to correct for the suppression to determine the non-suppressed magnitude and to determine if it does in fact look higher than CO$_2$ based parameterizations (when plotted with u). We calculated that the unsuppressed N00 would be 22% higher. However, we do not use this unsuppressed version of N00 for any calculations in this manuscript. It is an academic exercise. The same holds for the correction of the W14 parameterization. It is an application of our correction algorithm to calculate the slope of the W14 if one would correct for the gas transfer suppression that is within the data set.

**Figures 2 and 3 show that correcting for air flow separation moves wind speed to the left (i.e. Ualt <U10 when there's suppression), while k remains unchanged. Figure 6 (left panel) shows that correcting for air flow separation moves the global wind speed distribution to the left. Is it this adjustment in wind (rather than an adjustment to k) that causes the global fluxes to reduce in magnitude? But then Figure 4 and 5 show that the actual k values (the individual dual tracer points) are adjusted upwards (rather than the wind speeds adjusted leftwards) for the Nightingale et al 2000 data. How? Not clear.**

Answer: In relation to goal 2, we try to find a way to practically address the influence of gas transfer suppression. We think that when the wind speed u$_{10}$ gets close to the wave's phase speed (normally the wind speed is slower) gas transfer suppression happens. At the time before the onset of gas transfer suppression, the interfacial k behaves like the chosen linear k vs. u parameterization. Then wind speed increases and u$_{tr}$, in the wave's reference frame, gets smaller and gas transfer suppression sets in, decreasing k.

In our model we decrease u$_{10}$ and calculate the point at which gas transfer suppression sets in, u$_{alt}$. u$_{alt}$ is the wind speed with the maximum k possible at those conditions. As u$_{alt}$ does not appear in the physical world, it is an estimate of the wind speed at which the decrease/suppression of k begins and can be used in two ways:
a) If one has a global wind speed distribution, you can use u$_{alt}$ as the maximum wind speed for each data point before gas transfer suppression decreases k. This is what we have done to calculate the influence of gas transfer suppression on the Wanninkhof 2014 parameterization. That is why we moved the points on the wind speed axis and the global

distribution tends slightly towards smaller wind speeds.

b) If we have measured k values with a given wind speed we propose that the larger the difference between $u_{10}$ and $u_{alt}$ the larger the decrease of k, which we could call $\Delta k$. We compute the $\Delta k$ value with the DMS k vs u relationship:

$$\Delta k = (3.1 \cdot u_{10} - 5.37)\text{-} (3.1 \cdot u_{alt} - 5.37) = 3.1 \cdot (u_{10} - u_{alt})$$

Because we hypothesize that gas transfer suppression only affects the interfacial gas transfer, we use the DMS parameterization for correction. Correcting a data point using $\Delta k$ or moving it from u10 to ualt (with constant k) is equal, with regard to interfacial gas transfer only. For figures 2 and 3, we moved the points along the x-axis. For the N00 correction, we moved along the k axis, as the N00 parameterization has a significant bubble gas transfer contribution. For flux calculations, we find ualt along the x axis to obtain the unsuppressed k (i.e. $\Delta k$) (See manuscript Eq.).

**Seems to me that there are two possible philosophical approaches: a) U10 is a good predictor of k, and thus points below the expected mean k vs. U10 relationship (suppressed data) should be adjusted upwards in terms of k, or b) U10 is not a good predictor of k as waves are also important; thus we either need a new x variable that includes wave-wind interaction, or adjust U10 to account for waves as the authors have done in the Ualt calculation. But in this paper the authors seem to be taking both of these approaches.**

Answer: In our opinion, $u_{10}$ is a decent predictor of k and is especially useful with respect to the linear relationships for interfacial gas transfer. However, in regard to this comment, our approach is not philosophical, but rather practical. Wind speed is easily measurable by non-meteorologists and atmospheric scientists. As stated above, we try here to establish a first step to correct and estimate the influence of gas transfer suppression generally. Of course, given the spread in published wind speed based k parameterizations, we would ideally like to move beyond wind speed. To date, there are promising alternatives, but a parameterization with e.g. wave slope, friction velocity, would not be easy to use by a wide range of scientists.

**The bottom line is that the authors are trying to address an interesting and important topic. Unfortunately I find the paper far from publishable currently. And so I recommend a major revision and give the authors a chance to clearly address the issues raised.**

**Comments: At a given wind speed, there is probably a range of gas transfer velocities as a function of sea state. Recent works from Blomquist et al 2017 and Brumer et al 2017 demonstrate the efficacy of the 'wave-wind Reynolds number'. Some of the variability in sea state may be encapsulated by the authors' transformed Reynolds number, fine.**

Answer: We thank Dr. Yang for seeing that our formulation might be useful. Regarding the papers cited, we think that there are some points that need to be addressed further:

a) Blomquist et al page 8044: "However, a strong relationship between sea state and transfer velocity is not evident in this data set." As quoted they find no relationship between the general transfer velocity of DMS and $CO_2$ and sea state. However, they pick out the dates 24-25 October, when they find a dependency of only $CO_2$ (not DMS) k with sea-state. They state: "DMS transfer velocities show much less relationship to sea state development… with

scant evidence of significant enhancement or suppression in the presence of large waves." For $CO_2$ they state: "For example, by 12:00 on 26 October, wind speed had decreased to well below 15 ms$^{-1}$ but $k_{660}$ $CO_2$ remained significantly greater than 100 cmh$^{-1}$. This behavior is consistent with a low-solubility gas sensitive to the effects of breaking waves and bubble injection." As a consequence, they use the wind-wave Reynolds number to parameterize the gas exchange. In our opinion this is a parameterization for interfacial + bubble mediated gas transfer and takes the effect of sea state on breaking waves and bubble volume into account. We think the wind-wave Reynolds wave number is a valid and good description of the bubble effect on top of interfacial gas transfer. Our formulation for "gas transfer suppression" is affecting interfacial gas transfer only. In their data set no gas transfer suppression is evident and, therefore, their conclusion cannot be transferred to our model.

They also investigate previously published gas transfer data, which show signs of gas transfer suppression. They cannot explain the suppression using their wave-wind Reynolds number. This is plausible because, in our opinion, it specially focuses on the bubble part of the gas transfer. They also address the Soloviev 2007 parameterization and state: "The interfacial transfer model of Soloviev (2007) incorporates a wave age dependence that acts to reduce surface renewal and gas transfer in the presence of large waves, but these effects should apply equally to conditions on all these projects and on that basis does not provide a satisfying explanation of the observed differences." The Soloviev 2007 parameterization incorporates friction velocity and wave period (wave speed) which are good descriptions of wind-wave interactions. The downside here is that friction velocity does not account for the relative wind speed between wind and waves. In the paper " Bubble-Mediated Gas Transfer and Gas Transfer Suppression of DMS and $CO_2$" Zavarsky et al 2018, the DMS dataset and wind wave data of Hiwings is used to calculate the transformed Reynolds number. The transformed Reynolds number parameterization shows that there is no gas transfer suppression. This is in accordance with the findings of Blomquist et al 2018.

b) Brumer et al 2017 in the Keypoints section: "Wave-related Reynolds numbers provide a unique universal relationship for $CO_2$ gas transfer that transcends the quadratic-cubic conundrum."
Brumer et al 2017 also use the wave-wind Reynolds number. In our opinion this is again a description of the bubble mediated part of the gas transfer velocity. Figure 2 of their publication shows that for $CO_2$ the different measurement campaigns fall on top of each other using the wind-wave Reynolds number as parameter. In Figure 3, for DMS, this is not so much the case and clearly it can be seen that they cannot explain the gas transfer suppressed data points. They state: "One could a priori expect DMS to be less sea state-dependent than CO2 as its increased solubility means that its transfer velocity depends less on bubble-mediated transfer." If the sea-state dependence mostly relates to the bubble mediated gas transfer this statement is true. Generally, we think that DMS is more sea state dependent if it comes to gas transfer suppression, which is influencing the interfacial part of k. We can use our model to explain gas transfer suppression on various occasions (see Zavarsky et al 2018). However, Brumer et al say, "Weaker dependence on sea state may account for the increased scatter observed in the relationship between both the wave-wind and breaking Reynolds numbers and kDMS660. Sea state, represented as either the significant wave height or wave age, does not reconcile outliers in the SO GasEx and Knorr11 DMS data set." Again, they cannot explain gas transfer suppression.

**At expected times of gas transfer suppression, the authors decided to adjust the wind speed (U10) downwards to a transformed wind speed (Ualt) by using a threshold in the transformed Reynolds number. I think this binary treatment (i.e. either suppressed or not suppressed, instead of varying degrees of sea state effect) is overly simplistic**

Answer: Dr. Yang is right; we use a simplistic approach. As stated above, we wanted to provide an easy way for the larger community to correct for gas transfer suppression. Additionally, in aerodynamics (airflow over a wind, or airflow over a sphere), stall or no stall, attached or detached is a binary state. This might not be entirely true for waves and there are definitely transitions zones or hysteresis. There could be dependencies on surface roughness, wind speed, and wind direction, which gradually transfer the state from suppressed to non-suppressed and back. Right now, we think that this is far from measurable or addressable and we decided to start with what is firmly known.
For example, sea ice models still use the approach that under a threshold temperature the water surface is fully ice covered. We all know this is not precisely true, but it is a good start, on first principles, that will develop and get more sophisticated over time.
We added the following text to the manuscript introduction: "It is a binary view, but in aerodynamics stall conditions, flow detachment and reattachment are binary as well, so we adopted this view."

Our approach to describe a transition between a suppressed and an unsuppressed state is a statistical one. We calculate the amount of times, within the measurement or cruise time of days or weeks, when the wind-wave interaction was below or above the threshold. We can make a probability statement about the likelihood of seeing gas transfer suppression. In the paper Zavarsky et al 2018 you can see that this likelihood correlates with the decline of gas transfer velocity. You can also see this in the gas transfer suppression index in Figure 4 of this manuscript. The same problems (e.g. statistical vs episodic) exist between toxicologists and epidemiologists. One cannot certainly say that this very cigarette caused the disease or even put a number on its toxicity (although each cigarette poisons the body), but one can say that the likelihood of smoking x cigarettes over y years increases the likelihood of cancer by z%.

So far gas transfer suppression has not been explained at all (usually publications invoke hypotheses about the microlayer and wave shielding influences).  We make a first simple attempt to describe and parameterize this process using measured data that seems to hold in the majority of cases (Zavarsky et al 2018).

**What is the quantitative reasoning for adjusting wind speed downwards to the threshold REtr value in the case of suppression? Why not adjusting to an even lower |REtr| value, for example? And could there be times when k is 'enhanced' relative to the mean relationship? For DMS (Figures 2 and 3), k is simply shifted to the left due to the U10 to Ualt correction. However, Figure 5 shows that the dual tracer k values from Nightingale et al 2000 are actually shifted upwards, while wind speed remains unchanged. It looks like R2 is worse in Figure 5 than in Figure 4. How did the authors make this latter correction (Eq. 14?) and why the inconsistency in approach?**

Answer: In our model there are only two states: Suppressed and non-suppressed. We reduce the wind speed to the transition point between the suppressed und unsuppressed state. This wind speed relates, using a k vs u relationship, to the maximum k value possible

in this condition. As answered above, we have found that it is the same to adjust along u or along k for interfacial k. Since the dual tracer measurements have the additional influence of bubbles, we decided it is better to shift in k space, using only ko. In our opinion it is arbitrary to reduce the wind speed to a lower $Re_{tr}$. We reduce it to the transition point, as stated above in relation to the binary nature of the suppressed state. This is the maximum k possible for this wind-wave condition.

To date, there is not much evidence in the literature of k enhancement, so we do not address that here (we focus on suppression).

We stated the reason for moving along the k axis (y-axis) above. A worsening of $r^2$ value is a quality criterion, but not exclusively. Although it is true that the new fit has a worse RMSE (original 06.37, new quadratic only=10.6, new quadratic and linear=9.1), we do not think this is indicative of whether or not we deal with the suppression correctly. They could still be influenced e.g. by the presence of surfactants and different bubble terms.

**The authors did not apply the U10 to Ualt correction to the N00 data, as with DMS, because N00 data are more affected by bubbles? Also, the authors implied that the N00 dataset were taken in places (many coastal) and during times when gas transfer suppression is predicted to happen more often than the global average. Following that logic, shouldn't the global fluxes be higher, and not lower, if the original N00 contained a lot of suppressed gas transfer data?**

Answer: Dr. Yang is correct. We do not adjust to $u_{alt}$ because of the bubble effect. Since the suppression only acts on ko, we cannot "correct" N00 in the same way as for DMS. The terms must be separated.

As stated above, we think that the bubble term for $^3He/SF_6$ balances out the gas transfer suppression in ko. Therefore, it is suitable and used for $CO_2$ flux calculations and we do not need to correct the resulting fluxes to higher values. Also, as stated above, the N00 "correction" is an academic exercise, showing that suppression is ubiquitous and appears even in our commonly used parameterizations. When "corrected" we see the logical effect of a larger bubble term on the parameterization when compared with other parameterizations.

**In the case of W14, it is a single global average point averaged over multiple years. It presumably does include the full range of sea states. This single k point is pinned against a global mean wind speed (accounting for wind distribution). So if the functionality of W14 is correct, I don't see how it needs to be corrected at all to account for air flow separation. Does the right panel of Fig 6 imply an upward adjustment in the k value, or a leftward adjustment in wind speed? Shouldn't fluxes computed from [original W14 x NCEP wind speed distribution] be the same as those computed from [adjusted W14 x corrected wind speed distribution]? It's worth noting that in the revised wind distribution, there is far more occurrence of 'zero wind speed', which in the W14 formulation would result in zero flux. Are the authors saying that under conditions of moderate-to-high wind speed, when wind and waves follow each other, there is no gas transfer?**

Answer: The gas transfer velocity is set by $^{14}C$ measurements. So, to calculate the k vs u relationship, one needs the wind speed distribution. The underlying functional form to transfer the global transfer velocity to a wind speed parameterization is given by Wanninkhof 2014. We use the same formula (simple quadratic with no offset).

We say that $u_{alt}$ is the wind speed that actually acts on the ocean gas transfer in the case of gas transfer suppression. U10 is the wind speed one measures, but is not relevant in the suppression environment. As a consequence, we use a globally calculated $u_{alt}$ distribution to calculate the theoretical k vs u relationship from the [14]C inventory without any suppression. Again, this is an academic exercise and is only used to compute the flux difference between the unsuppressed case (hypothetical) and the average suppression case (using the normal W14 parameterization). The occurrence of more zero wind speeds is apparent. In our calculation, zero wind speed means no gas transfer. We know that there are doubts that zero wind speed means zero flux. However, W92 and W14 use a formula that implies this. We believe gas transfer suppression occurs at all wind speeds. A suppression at low wind speed, has so far, not been observed, but using the model it is possible that, for example, a u10 of 3 ms$^{-1}$ gets reduced to a $u_{alt}$ of 0 ms$^{-1}$.

**Some technical comments: Your Eq. 1 is presented from the perspective of air concentrations. Since you're talking about water-side controlled gases, it seems more appropriate to present this Eq. from the perspective of water concentrations, i.e. k * (Ca*H - Cw) or k * (Ca/H - Cw), depending on whether your H is water to air or air to water. Also, your Eq. 1 adopts the convention of positive flux into the ocean. That's consistent in sign to your global CO2 flux, but not to your DMS flux. Please be consistent.**

Answer: We changed the Equation (1) to:

F=k*(-Ca/H).

We use the convention of positive flux out of the ocean and negative flux out of the ocean. As a consequence, we also change the order in Equation (2).

**Eq. 3: I think you have left out the H term. Should be 1/ktot = 1/kw + H/ka (if H is water to air) In many of the plots, I think it's misleading to call Ualt 'wind speed' on the x-axis and have both k vs U10 and k vs Ualt on the same plot. Figures 9 and 10. Which original parameterization is used? Please specify in the captions.**

Answer:  We changed Eq. 3 accordingly.

We do not think it is necessary to make changes according to the label of the x-axis. All x-axes have the label "wind speed". It is stated in the legend and the caption which data set (u10 or $u_{alt}$; also, $u_{alt}$ is u10) is plotted.

Figures 9 and 10 show the difference between suppressed and unsuppressed gas transfer. The difference is calculated using the bulk formula F=k*ΔC and using a change in k according to our model:
$$\Delta k = (3.1 \cdot u_{10} - 5.37) \text{-} (3.1 \cdot u_{alt} - 5.37) = 3.1 \cdot (u_{10} - u_{alt})$$

There is no original parameterization, from which we subtracted Δk. We used the ZA18 to correct for the gas transfer suppression and therefore for the calculations of the change of flux in Figures 9 and 10.

**Finally, the results from ZA18 are heavily used in this current paper. While the presented argument of air flow separation is a neat theory, I don't think it's well**

**backed up by the observations for at least three reasons: 1) It is conceivable that transfer velocity varies with the directional difference between wind and wave as well as with the relative wind velocity relative to the wave phase speed. However, I don't understand the directional dependence in the formulation of the transformed Reynolds number. Air flow separation and sheltering are argued to occur when wind and waves are aligned (and not occur when they are orthogonal). However, cos (0) = 1 and cos (90) = 0. And it is a low transformed Reynolds number that is argued to cause a suppression (or limitation) in gas transfer. This seems contradictory.**

Although, as Dr. Yang states, these comments are not about the current paper, but about a previously published paper, we will still respond below.

Sheltering and gas transfer suppression are two different processes. We discuss gas transfer suppression due to a flow detachment in the paper and only mention the concept of sheltering as a previously hypothesized reason for suppression.

Our model of gas transfer suppression is based on the transformed Reynolds number $Re_{tr} = \frac{u_{tr} \cdot H_S}{\vartheta} \cdot cos(\theta)$. The number depends on the relative wind speed $u_{tr}$, the wave height and the angle of attack Θ. Generally, gas transfer suppression can occur at all angle of attacks if Retr drops below the threshold. At angles close to Θ=90° gas transfer suppression should not occur and, in fact, we think this is hardly measurable and sparsely occurring.
The reason for normal gas transfer is that the object in the flow path (wave) creates turbulence, which counteracts the flow separation. This turbulence can be created in two ways (1) larger $u_{tr}$, (2) higher waves. Both increase $Re_{tr}$, which leads above the threshold to normal gas transfer.
The factor cos(Θ) basically describes the wave slope. A wave with a certain slope creates according to the wave height turbulence and if $Re_{tr}$ is above the threshold normal gas transfer occurs. If the flow experiences an angle of attack (cos(Θ)), the slope of the wave changes. A flatter slope results in less turbulence to counteract the flow separation. Hence one needs a higher obstacle or larger $u_{tr}$ to counteract the flow separation. This is the reason why cos(Θ) is a factor in the $Re_{tr}$ formula.

We added an extra sentence to the section 2.4 and added a graphical illustration to the supplemental material.

**2) ZA18 explains previous transfer velocity datasets with the flow separation theory, but did not use actual (in situ or modeled) wave data. This is a significant shortcoming in my view. I have the ECMWF wave and in situ wind data from those cruises. It is not obvious that waves from cruises when gas transfer suppression were observed differed obviously from the waves during other cruises. The authors are welcomed to contact me and use these data to further improve their work**.

We did use modelled wave data at the times of the cruise. The cruise positions and times are available online. We then used the WWIII model hindcast to get the wave data for that specific cruise track, at the specific position and time of the cruise.
The new concept includes a Galilean transformation of the wind into the wave's reference system and then a calculation of the transformed Reynolds number. A comparison between different cruises regarding waves only is not sufficient. It is necessary to know and compare the wave speed, wave direction, wind speed, wind direction, and wave height. A description of the sea state with, for example, cp/u is also not enough, as only absolute values are

considered. A description using absolute values might lead to similar results, but not in all cases.

**3) a lot of high points and noise in the kDMS and kCO₂ data occurred when the delta C were very small. Not only are fluxes very noisy under these conditions, any small bias in delta C would also significantly affect the derived k. Is there still a noticeable 'suppression' if the authors remove these low delta C points?**

For DMS we removed all points with a seawater concentration lower than 2 nM/l, for $CO_2$ we removed all point with a delta C lower than 20 μatm. The results are shown in the two figures below. The binned data line in both figures is based on the entire dataset. We see that even omitting points of low DMS seawater concentration or low delta $pCO_2$, gas transfer suppression remains in the dataset.

[Figure]

[Figure]

These last comments are not criticisms of the ACPD paper, but partly explain why I find the current paper rather unconvincing.

[revised manuscript text omitted]

---

## Author Response (AR2)

Co-Editor Decision: Publish subject to minor revisions (review by editor) (03 Dec 2018) by Martin Heimann
Comments to the Author:
Please revise your manuscript according to the thorough recommendations by the reviewer given below. A final review before publishing will be done by me as editor.

Review comments:

I find the revised version more digestible but I'm still unclear on a few aspects. Please see below:

**I don't think the words 'corrected' and 'corrections' are the best choices for describing gas transfer suppression and associated adjustments. maybe 'adjusted' or 'unsuppressed' instead.**
**Please be precise in the language. For example, captions for Figs 2 and 3 should read "adjustment to the k v U relationship", rather than "correction of fluxes"**

We changed the wording from correct to adjust. You are right in Figs. 2 and 3 k vs u relationships are shown. We changed the caption.

**Eq 1 and 2 still from the perspective of air concentrations. i.e. k shown = Ka, not Kw should be flux = k(Cw - Ca*H) when k = Kw**
**Eq 3. earlier k is used, and now ktotal is used. please be consistent. I suggest Kw or Kt for Eq. 1-3**

We changed all ks to ktot and changed Eqs 1 and 2.

**top of p. 3. the dual tracer (3He/SF6) method likely doesn't capture the full extent of bubble-mediated gas transfer. see: Asher and Wanninkhof 1998, vol103, 10555-10560, JGR. accounting for bubbles will likely increase the N00 curve**

You are right, but Asher and Wanninkhof only refer to a comparison of dual tracer data. In the conclusion they state:
"The effects of bubble transfer are much larger when calculating transfer velocities for $CO_2$ from transfer velocities measured for 3He. In this case, neglecting bubbles causes the transfer velocity for $CO_2$ at a wind speed of 17 m $s^{-1}$ to be overestimated by 23%. Overall, bubble-mediated exchange will be important in correcting k values between two gases when there is a large difference in their solubilities and Schmidt numbers."

Our correction leads to a 22% increase, which closely matches this estimation. We added a sentence to page 11 line 31 in the new text:
"Asher and Wanninkhof (1998) state that $SF_6/^3He$ gas transfer measurements could lead to a 23% overestimation of $CO_2$ gas transfer velocities. After adjusting N00 for gas transfer suppression, the difference between fluxes computed using the adjusted and original N00 closely matches this estimation."

We think there is a problem when using dual tracer k values ($SF_6/^3He$) for other gases with different solubilities, like $CO_2$. This what many people in the community do when using the N00 k parameterization and applying the Schmidt number correction. We address this problem in this manuscript.

**p. 4, line 12. 17% reduction due to inclusion of DMS air concentration seems large. 100 ppt of DMS in air converts to an equilibrium concentration 0.06 nM at 20 deg C, while a typical seawater DMS concentration is about 1.5-2 nM**

Here we cite the global modeling study of Lennartz et al. (2015). They found a 17% reduction in computed fluxes when including an interactive atmosphere in the flux calculations as compared to Lana et al (2011), which sets the atmospheric values to 0. Therefore, we do not mean to say that each cruise will have a 17% reduction when the atmosphere is included, but we want to caution that this approach could lead to overestimated fluxes.

**p. 5, line 29. so this assumes that N00, W14, and TAK09 aren't suppressed already**

At this point in the manuscript, we describe our theory and assume that the published parameterizations are not suppressed. Then we find gas transfer suppression at the times the data points were recorded. We hypothesize that W14 and TAK09 already have a mean global gas transfer suppression built into them, as they use global data sets to calculate k.

**p. 6, line 24. the two approaches are equivalent in the case that they result in the same k vs. u slope. but of course they will result in two different mean K values, for example**

We added a few words on p. 6 line 27 to clarify this point:
*"The shift along the x-axis is equivalent to an addition of deltak, for a given k vs. u relationship,  to balance gas transfer suppression (see appendix)."*

**p. 7, line 26. "we interpolated the wind and wave data to 1 h time steps..." so the authors used the 2014 wave information, and not modelled wave data from the actual year of the dual tracer measurements, correct? are the modelled wave data taken from the same locations as the actual dual tracer measurements? are the "1 h time steps" around the actual times of the dual tracer measurements? or they're 1 h time steps of the entire year?**

For the analysis of N00, W14 and the Knorr11 data we used data at the specific time and location of the measurements. We clarified this now in the methods section p3 line 28 of the new version:
*"The model is calculated for the global ocean surface excluding ice covered areas with a temporal resolution of 3 h and a spatial resolution of 0.5° x 0.5°. The data for the specific analysis of the N00, W14 parameterizations and the Knorr11 cruise (sections 4.1, 4.2, 4.3) was obtained from the model for the specific locations and times of the measurements."*

We also clarified this in p7 line 32 of the new version:

*"As the measurement time of the dual tracer technique is on the order of days, we interpolated the wind and wave data, obtained from the WWIII model for the specific time and location, to 1h time steps and calculated the number of gas transfer suppressing and gas transfer non-suppressing instances."*

**p. 10. line 18. however a gas transfer 'suppression' was observed for the Southern Ocean GasEx cruise**

You are right. We changed the wording to "with less". There is of course gas transfer suppression in the Southern Ocean, but according to our calculations, it is less likely.

**p. 10, line 21. Most critically, this is where I'm lost. the authors already show that the original N00 is heavily suppressed (by "22%"). So why is the subtraction of deltaK in Eq. 14 necessary in the global estimates?**

In our opinion, the original N00 parametrization describes the unsuppressed gas transfer of $CO_2$. However, this is just coincidental. We agree with Asher and Wanninkhoff (1998, JGR) that there will be a lot of overestimation (due to bubbles) when calculating $CO_2$ gas transfer velocities from $^3He/SF_6$ velocities. Since we show that N00 is heavily gas transfer suppressed, we hypothesize that this compensates for the bubble-mediated transfer. Therefore, N00 (heavily suppressed $^3He/SF_6$ data) can be used as a $CO_2$ parameterization fur unsuppressed conditions. To calculate suppressed $CO_2$ gas transfer velocities, we have to suppress N00 again.

For clarity, we removed the global $CO_2$ flux calculation and simply estimate the change in $CO_2$ fluxes due to gas transfer suppression.

**If anything, it seems that if the global average k suppression is 10%, and N00 is suppressed by 22%, one should get a HIGHER, not LOWER global flux when using the adjusted N00 relationship (compared to using the original N00 relationship)**

To compute the global DMS  fluxes compared to Lana et al 2011, we used the original N00 parameterization. Lana et al 2011 also used it, not knowing about gas transfer suppression and assuming that N00 is not suppressed.  Then we apply our suppression algorithm to N00 (because Lana et al 2011 assumed it is unsuppressed) and compute new DMS fluxes. These series of adjustments leads to lower global DMS fluxes. We added the following text to the new manuscript section 4.4:

*"The calculated total emission from the original N00 parameterization is 50.72 Tg DMS y-1 for the year 2014. Then we use our estimations of ualt and Eq. 14 to subtract gas transfer suppression from the original N00 parameterization. The resulting reduced total emission is 45.47 Tg  DMS yr-1, which is a reduction of 11%."*

**The Tak09 relationship, similar to W14, is based on multiple years of observations and presumably includes the effect of any gas transfer suppression already.**
**The authors state the following and I think the same applies to TAK09.**

**W14 parameterization is just calculated for comparative and example reasons to show the effect on this parameterization type. The W14 parameterization already includes an average estimate for gas transfer suppression and no suppression needs to be added on top.**
**I don't think it makes sense to apply the gas transfer suppression model to W14 and TAK09 and get revised global CO2 fluxes. I think those values should be removed from** **Table 3.**

We removed the global $CO_2$ fluxes and just estimate the flux change for $CO_2$ on a monthly, local, datapoint by datapoint level. We also state that W14 and TAK09 already include gas transfer suppression and should be used for global studies, as they more realistically represent all the physical processes influencing gas transfer.

[revised manuscript text omitted]

---

## Author Response (AR3)

Dear Editor,

thank you very much for your corrections and suggestions.
We made the necessary changes in the abstract. We also added a small paragraph in the introduction to explain the aerodynamical model of the flow around a sphere. We also corrected some sentences throughout the manuscript. These corrections should increase the readability and do not change the scientific content.

We added the new manuscript as well as a tracked-change manuscript.

[revised manuscript text omitted]